# Unveiling Therapeutic Potential: Targeting *Fusobacterium nucleatum*’s Lipopolysaccharide Biosynthesis for Endodontic Infections—An In Silico Screening Study

**DOI:** 10.3390/ijms25084239

**Published:** 2024-04-11

**Authors:** Nezar Boreak, Ethar Awad Alrajab, Rayan Ali Nahari, Loay Ebrahim Najmi, Muhannad Ali Masmali, Atiah Abdulrahman Ghawi, Mohammed M. Al Moaleem, Majed Yahya Alhazmi, Abdulrahman Abdullah Maqbul

**Affiliations:** College of Dentistry, Jazan University, Jazan 45142, Saudi Arabia; ethar8270@gmail.com (E.A.A.); rayannaharii@gmail.com (R.A.N.); loaynajmi125@gmail.com (L.E.N.); mohanedd19999@gmail.com (M.A.M.); atiy07sa@outlook.sa (A.A.G.); malmoaleem@jazanu.edu.sa (M.M.A.M.); majedalhazmi18@gmail.com (M.Y.A.); abdulrhman.maqbul@gmail.com (A.A.M.)

**Keywords:** endodontic infection, pulpitis, *F. nucleatum*, lipid A, acyltransferases, combinatorial libraries

## Abstract

Complex microbial communities have been reported to be involved in endodontic infections. The microorganisms invade the dental pulp leading to pulpitis and initiating pulp inflammation. *Fusobacterium nucleatum* is a dominant bacterium implicated in both primary and secondary endodontic infections. Drugs targeting the molecular machinery of *F. nucleatum* will minimize pulp infection. LpxA and LpxD are early acyltransferases involved in the formation of lipid A, a major component of bacterial membranes. The identification of leads which exhibit preference towards successive enzymes in a single pathway can also prevent the development of bacterial resistance. A stringent screening strategy utilizing physicochemical and pharmacokinetic parameters along with a virtual screening approach identified two compounds, Lomefloxacin and Enoxacin, with good binding affinity towards the early acyltransferases LpxA and LpxD. Lomefloxacin and Enoxacin, members of the fluoroquinolone antibiotic class, exhibit wide-ranging activity against diverse bacterial strains. Nevertheless, their effectiveness in the context of endodontic treatment requires further investigation. This study explored the potential of Lomefloxacin and Enoxacin to manage endodontic infections via computational analysis. Moreover, the compounds identified herein serve as a foundation for devising novel combinatorial libraries with enhanced efficacy for endodontic therapeutic strategies.

## 1. Introduction

Endodontic infections are inflammatory diseases caused by microbial communities located in the root canal. Endodontic infections are classified as primary and secondary infections based on the time of infection [1]. Primary infection is caused when the pulp is invaded and colonized by the oral microbes, while secondary infections are due to persistent microbial root canal infection following endodontic treatment. Secondary infections can also be due to the introduction or retention of microbes during the treatment of the primary infection [2].

The microbial diversity of these infections can vary depending on various factors such as the stage of infection, host immune response, and previous treatment history. Several studies have demonstrated the polymicrobial nature of endodontic infections through microbiological analysis techniques such as culture-based methods, polymerase chain reaction (PCR), and next-generation sequencing (NGS) [1,2,3]. The polymicrobial communities, enclosing Gram-positive and Gram-negative facultative bacteria and anaerobes, interact with each other, forming complex microbial inter-relationships. The core microbiome encompasses the main candidate endodontic pathogens: Gram-positive bacteria (*Actinomyces* species, *Streptococcus* species, *Propionibacterium* species, and *Cutibacterium acnes*) and Gram-negative bacteria (*Fusobacterium nucleatum*, *Dialister* species, *Porphyromonas endodontalis*, *Porphyromonas gingivalis*, *Prevotella* species, *Tannerella forsythia*, *Treponema* species, *Parvimonas micra*, *Filifactor alocis*, *Pseudoramibacter alactolyticus*, and *Olsenella uli*) [4]. The presence of multiple bacterial species in endodontic infections highlights the complexity of these infections and underscores the importance of comprehensive microbial identification and targeted antimicrobial therapy for successful treatment outcomes. The primary endodontic infection is mostly composed of Gram-negative bacterial species [4]. Furthermore, the number of bacteria contributing to primary endodontic infections is higher when compared to secondary infections [5].

The relevance of antibiotic resistance from the usage in periodontal infections extends to the management of endodontic infections [6,7]. Shared microbial communities between the two conditions suggest that antibiotic resistance observed in periodontal infections could affect endodontic cases. Therefore, it is essential to exercise careful antibiotic prescription practices to mitigate this risk. Although antibiotics can aid in endodontic infection management, their use must be carefully balanced to combat resistance.

*Fusobacterium nucleatum*, a Gram-negative anaerobe, is ubiquitous in the oral cavity [8]. *F. nucleatum* is absent or infrequently detected in other parts of the body under normal conditions. The bacterium is frequently associated with endodontic infections, which includes pulp necrosis and periapical periodontitis. The frequency of *F. nucleatum* rises in tandem with the severity of the disease, the advancement of inflammation, and the depth of pockets [9,10].

The frequency of *Fusobacterium nucleatum* in endodontic infections provide valuable insights into microbial composition and pathogenicity of these infections, influencing diagnostic and therapeutic strategies [9,11,12]. *F. nucleatum*’s connection with endodontic infection necessitates the identification of new therapeutic molecules targeting the bacterium.

The Raetz pathway of lipid A biosynthesis plays a vital role in the survival and fitness of Gram-negative bacteria. Constitutive biosynthesis of lipid A is required for the viability of nearly all Gram-negative bacteria. Most Gram-negative bacteria share a distinguishing feature: they possess a lipopolysaccharide (LPS) layer on their outer membrane. LPS molecules comprise three distinct structural components, commencing with lipid A, which binds to the outer membrane’s phospholipids. Subsequently, a core oligosaccharide follows, and finally, the O antigen. This LPS layer serves as a protective barrier against the external environment and imparts structural stability to the outer membrane, primarily due to the densely packed structure influenced by the hydrophobic properties of lipid A. Consequently, it plays a significant role in the regulation of molecules crossing the bacterial membrane. Given the contrasting natures of its structural constituents, it prohibits the passage of both hydrophilic and hydrophobic molecules through the membrane [10].

The initial phase of LPS layer biosynthesis involves the creation of lipid A molecules within the cytoplasm. This process takes place with N-acetyl glucosamine as the precursor, coupled with a UDP molecule (UDP-GlcNAc). LpxA initiates this by acetylating the precursor, followed by deacetylation by LpxC. Subsequently, LpxD acetylates the resulting product again, functioning similarly to LpxA. Both LpxA and LpxD are indispensable enzymes. The Lpx family of proteins assumes regulatory control over this pathway, ultimately culminating in the formation of the lipid A molecule [9].

While LpxC lacks homology with LpxA and LpxD, the latter two enzymes exhibit numerous distinct structural characteristics that align with their functional similarities in facilitating the transfer of a 10 or 12 carbon chain fatty acid from ACP to UDP-GlcNAc through a coordinated acid-base mechanism [13]. LpxA and LpxD form biological homotrimers. Though *F. nucleatum*’s LpxA and LpxD shares only 26% sequence identity, they demonstrate a remarkable conservation of protein backbone and side chain structural features. Inhibitor discovery against LpxA and LpxD remains largely unexplored. The distinct yet mutually shared structural resemblances of LpxA and LpxD makes them suitable for dual-targeting inhibitor design. The main advantage of the dual-target inhibitor design is that it can increase potency while minimizing the chances of resistance development.

In the current study, a computer-aided ligand-based virtual screening approach was utilized to identify promising lead molecules against LpxA and LpxD of *F. nucleatum* based on the known inhibitors of *P. aeruginosa*. Physicochemical and ADME-T analyses of the lead compounds were carried out to investigate the distribution of their properties. The best-binding lead compounds identified after initial screening were subjected to molecular dynamics and simulation to analyse the conformational dynamics of the lead compounds when bound to LpxA and LpxD, respectively.

## 2. Results

### 2.1. Identification of Inhibitors against Lipid A Biosynthetic Acyl Transferases

By exploiting the rule of ligand chemical similarity (compounds with similar structures are likely to have similar activity) and utilizing a hybrid virtual screening (VS) approach combining both structure and ligand-based VS strategies, therapeutic leads against the early acyl transferases LpxA and LpxD were identified.

The compound collection from the PubChem and ChEMBL databases identified a total of 1930 compounds. After initial filtering by manual verification, around 590 compounds were retained and subjected to the VS approach (Appendix A). The compounds binding to acyl transferases were assessed based on their binding energy. The binding energy of compounds bound to LpxA ranged between −9.8 and −5.2 kcal/mol. Similarly, the binding energy of compounds bound to LpxD ranged between −14.4 and −5.9 kcal/mol.

### 2.2. Toxicity and ADME Screening

All the small molecules were screened in DataWarrior to screen them for Lipinski’s rule of five (RO5) and drug likeness. In addition, toxicity parameters such as irritating effects, mutagenicity, carcinogenicity, reproductive effects, and irritants were also computed. Out of 590 compounds, 178 passed RO5 with no toxicity parameters associated with them (Appendix A). All these compounds were further assessed for ADME parameters, which includes blood–brain barrier (BBB) permeability, gastrointestinal (GI) absorption, aqueous solubility, consensus octanol-water partition coefficient (cLogP < 5), topological polar surface area (TPSA < 140 Å^2^), Cytochrome P450 (CYP) inhibition, and PAINS (pan assay interference compounds). A total of 129 compounds passed ADMET filters and were chosen for further processing (Appendix A). The binding energies of all 129 compounds for LpxA and LpxD were further assessed. Among the 129 compounds, Lomefloxacin (CHEMBL561) showed a high preference for LpxA with a binding energy of −8.4 kcal/mol. Similarly, Enoxacin (CHEMBL826) showed a high preference for LpxD with a binding energy of −9.7 kcal/mol. Both the compounds were tested for their biological activity spectrum using the PASS server. The toxicity and RO5 statistics for Lomefloxacin and Enoxacin are shown in Table 1 and Table 2.

Lomefloxacin was predicted to show anti-bacterial and anti-infective properties. Enoxacin showed an antibiotic Naphthyridine-like property along with anti-bacterial activity (Table 3).

Lomefloxacin and Enoxacin’s antibacterial efficacy was additionally confirmed through examination of the published literature. Lomefloxacin and Enoxacin are oral fluoroquinolone antibiotics that have wider bactericidal activity and oral bioavailability. The drugs also possessed good tissue penetration with favourable safety and tolerability profiles [14]. Lomefloxacin exhibits antibacterial activity against a wide range of Gram-negative bacteria [15,16]. Similarly, Enoxacin also exhibits activity against Gram-negative bacteria, but shows comparatively lower efficacy against Gram-positive microorganisms [17,18]. The comprehensive examination of ADMET properties and binding energy analysis indicated that acyl transferases LpxA and LpxD exhibited a particular preference for fluoroquinolones.

### 2.3. Binding Mode Analysis of Small Molecules with Acyl Transferases

Lomefloxacin showed a binding energy of −8.4 kcal/mol with LpxA. Lomefloxacin forms conventional hydrogen bonds with GLN69 of the A chain and GLY168 of the B chain. The drug also formed hydrophobic interactions with THR135 of the A chain and ASN132 of chain B. Salt bridges were also observed with HIS117 and HIS120 of the A chain (Figure 1). The pairwise sequence alignment between LpxA of *F. nucleatum* and *P. aeruginosa* revealed that the interacting residues were highly conserved between the species.

Enoxacin binds with a binding energy of −9.7 kcal/mol with LpxD. Enoxacin formed conventional hydrogen bonds with SER247 of chain A and VAL269 of chain C. Hydrophobic interactions were observed between ILE246, ALA264, and ALA283 of chain A and ILE252 of chain C (Figure 2). Based on the LpxD pairwise sequence analysis of *F. nucleatum* and *P. aeruginosa*, it was observed that the interacting residues were partially conserved between the species.

In summary, the docking studies indicated that both Lomefloxacin and Enoxacin exhibited strong binding to their respective targets, LpxA and LpxD, with favourable binding energies. Furthermore, they emphasize the presence of conserved residues in mediating interactions between drugs and their targets across diverse bacterial species.

### 2.4. Molecular Dynamics and Simulation Analysis

Molecular dynamics and simulation analysis were conducted to obtain a better insight into protein–ligand binding dynamics. Structural parameters such as RMSD, RMSF, Rg, essential dynamics, SASA, and timescale analysis were compared between the apoproteins and ligand-bound protein complexes. In addition, the number of intermolecular hydrogen bonds and the free energy of binding, along with various interaction energies, were analysed for the protein–ligand complexes.

#### 2.4.1. Root Mean Square Deviation (RMSD)

The structural stability of the apoproteins and ligand-bound protein complexes was monitored using RMSD calculations.

The comparison of apo-LpxA and ligand-bound LpxA protein RMSDs revealed that the deviations observed in the protein backbone in both systems were very close. Smaller deviations were higher in the complex LpxA when compared with the apoprotein. Both the systems maintained small drifts throughout the entire simulation time. However, the RMSDs were maintained below 0.3 nm throughout the simulation time. A closer analysis indicated that binding the ligand has decreased the deviation of the protein backbone to a lesser extent after 75 ns Figure 3a. The comparison of the RMSDs of apo-LpxD and complex LpxD revealed that the complex RMSD was lesser than the apo-state indicating that binding of the ligand has reduced the backbone structural movement of the protein Figure 3b. In addition, the ligand binding reduced the RMSD approximately from 0.5 nm in apoprotein to 0.2 nm in a complex state. In both the protein systems, the ligands adopt different approaches to interacting with the protein.

#### 2.4.2. Root Mean Square Fluctuations (RMSFs)

RMSF indicates the flexibility of the amino acid residues of the protein during the simulation. The comparison of residual fluctuations between apo-LpxA and complex LpxA revealed that residual fluctuations were higher in the complex system when compared to the apo state. Higher fluctuations were observed not only in the loop regions, but also in the chains where the ligand was bound. In the LpxD system, the residual fluctuations were higher in the apo state compared to the complex state. On closer analysis, irrespective of the loops and chains, the entire residual fluctuations were minimal in the complex state (Figure 4).

#### 2.4.3. Radius of Gyration (Rg)

Rg indicates the compactness and folding properties of the protein structures throughout the simulation time period. The comparison of the apoproteins with the complex proteins indicated that in both protein systems, the Rg value is smaller in the ligand bound proteins. The Rg value of Lomefloxacin-bound LpxA was maintained between 2.7 nm and 2.8 nm throughout the simulation time. The Enoxacin-bound LpxD showed an initial increase and converged to 3.05 nm at the later part of the simulation time period (Figure 5).

#### 2.4.4. Essential Dynamics

Principal component analysis (PCA) is a method for reducing dimensionality by analysing the variations in Cα atom positions. It constructs a covariance matrix, from which eigenvalues and eigenvectors are derived through diagonalization. These eigenvectors, known as principal components, are then projected onto an essential subspace, with each point representing a distinct conformation of a protein structure. The comparison of the first two principal components of the apoproteins with the ligand-bound protein complexes indicated that the ligand-bound forms covered a lesser region in the conformational space of the two principal components (PC1 and PC2) than the apo form. Upon ligand binding, the proteins were stabilized and occupied a small area in the conformational space, indicating the loss of flexible movements of the proteins (Figure 6).

#### 2.4.5. Solvent Accessible Surface Area (SASA)

SASA shows the interaction of the protein with the solvent molecules, which subsequently indicates changes in the protein conformation. The apo-LpxA and ligand-bound LpxA complexes showed only minor differences in the SASA values upon comparison, indicating that ligand binding does not significantly alter the conformation of the protein. In contrast, the SASA values of the apo-LpxD and ligand-bound LpxD complex showed significant changes. The solvent accessible surface area was significantly reduced in the LpxD protein upon ligand binding, which is evident from Figure 7.

#### 2.4.6. Intermolecular Hydrogen Bonds and Time Scale Analysis

Hydrogen bonding interactions formed between residues located in the active site of the protein and the ligand molecule are important to understanding the binding affinity of the molecules towards a drug target. The hydrogen bond analysis revealed that both the protein–ligand complexes maintained a minimum of one and a maximum of four hydrogen bonds throughout the simulation time period. The LpxA–ligand complex maintained an average of one hydrogen bond throughout the simulation time. The LpxD–ligand complex maintained an average of three hydrogen bonds during the simulation time (Figure 8).

In order to understand the structural and conformational changes to the initial state of the protein (before MD simulation) and the optimized state of the protein (during and after MD simulation), a comparative analysis by structural superposition of the structures using PyMOL was carried out. The protein–ligand complex snapshots at the 0 ns, 50 ns, and 100 ns time periods were retrieved and compared. The LpxA–ligand complex showed an RMSD of 0.129 nm at 50 ns and 0.136 nm at 100 ns from the initial starting structure. The LpxD–ligand complex showed an RMSD of 0.397 nm at 50 ns and 0.369 nm at 100 ns (Figure 9 and Figure 10).

#### 2.4.7. Binding-Free Energy Calculations

The interaction free energies of the ligand-bound protein complexes were calculated using the molecular mechanics Poisson–Boltzmann surface area (MM-BSA) approach. A stable trajectory was extracted from 80 to 100 ns of the global 100 ns and analysed. All energy terms and the total binding-free energies for these systems are shown in Table 4.

The MM-PBSA analysis indicated that van der Waal energy was highly favoured in both protein–ligand complex systems. The lower SASA energy value in both complex systems indicated that the protein structures were highly compact, allowing only a few solvent interactions.

The overall MDS analysis of the apoproteins and protein–ligand complexes suggested that both Lomefloxacin and Enoxacin exhibited strong binding to their respective targets, LpxA and LpxD, with favourable binding energies. Ligand binding reduces protein backbone movement, promotes compactness, and stabilizes the protein structures. Additionally, ligand-bound complexes occupy a smaller conformational space, indicating reduced flexibility compared to apoproteins.

The inappropriate use of antibiotics can lead not only to increased adverse events and healthcare costs but also to the risk of developing resistance [19]. While Lomefloxacin and Enoxacin may exhibit effectiveness in treating periodontal diseases, monitoring for resistance development is essential to ensure continued therapeutic efficacy. The emergence of resistance to Lomefloxacin and Enoxacin in periodontal diseases is a plausible concern. Continuous surveillance is crucial to ensure the sustained effectiveness of these antibiotics.

## 3. Discussion

The rise of bacterial pathogens that are resistant to major classes of commercially available antibiotics has generated a pressing demand for novel antibacterial agents. Recent genetic investigations have indicated that targeting the biosynthesis of the Lipid A anchor in the lipopolysaccharide (LPS) of Gram-negative bacteria represents a promising strategy [20]. In, the present work, combined molecular modelling and simulation techniques have been applied to screen multi-target inhibitors by checking their physicochemical properties and binding energies.

The advantage of utilizing ADMET filters is to restrict candidates which shows high potential of becoming drugs. Typically, due to the vast quantity of small molecules subjected to virtual screening, only the highest-ranked candidates, constituting the top 10% or less based on their binding energy, were deemed eligible for subsequent evaluation [21,22]. Through rigorous screening, Lomefloxacin and Enoxacin emerged as promising compounds due to their high affinity towards specific target proteins, LpxA and LpxD, respectively.

Subsequent molecular dynamics simulations provided detailed insights into the interactions between these compounds and their target proteins. The analysis revealed that Lomefloxacin and Enoxacin binding stabilized the protein–ligand complexes, resulting in reduced protein conformational movements and increased overall stability. Moreover, the compounds induced favourable changes in protein dynamics, such as decreasing residual fluctuations and promoting protein compactness.

The present study indicates the potential effectiveness of employing Lomefloxacin and Enoxacin to treat endodontic infections. The use of antibiotics like Lomefloxacin and Enoxacin could play a crucial role in particular clinical situations. Studies have suggested that Lomefloxacin and Enoxacin exhibit effectiveness against certain pathogens commonly associated with endodontic infections such as *Enterococcus faecalis* and other anaerobic bacteria [23,24]. Unlike periodontal infections, where antibiotic resistance is a growing concern due to frequent use, endodontic infections typically involve a confined space within the tooth, potentially allowing for more targeted antibiotic action. Therefore, Lomefloxacin and Enoxacin may offer a viable treatment option for endodontic infections, potentially minimizing the risk of antibiotic resistance associated with their use in other dental conditions. However, further clinical studies are needed to fully substantiate their efficacy and safety in this specific context. Determining the most suitable routes of administration for Lomefloxacin and Enoxacin in treating endodontic infections requires further investigation. Factors such as bioavailability, patient comfort, and clinical efficacy need to be considered before recommending specific administration routes [25].

These antibiotics hold importance due to their broad-spectrum activity against a wide range of bacteria, including both Gram-positive and Gram-negative strains. Their usage may be particularly relevant in cases where endodontic infections exhibit systemic involvement or fail to respond adequately to conventional treatment approaches. By incorporating Lomefloxacin and Enoxacin as an adjunctive therapy alongside mechanical debridement and irrigation of the root canal system, clinicians can enhance bacterial elimination and reduce the risk of treatment failure or systemic complications. However, it is crucial to emphasize the cautious use of these antibiotics, prescribing them for the shortest effective duration and closely monitoring patients for signs of treatment response and adverse reactions. Additionally, patient education on adherence and antibiotic resistance risks should underscore the importance of prudent usage of Lomefloxacin and Enoxacin in endodontic management, ensuring their efficacy while mitigating the risk of promoting antibiotic resistance. In general, integrating Lomefloxacin and Enoxacin strategically can substantially contribute to attaining favourable results in addressing intricate endodontic infections.

## 4. Materials and Methods

### 4.1. Proteins Structure Preparation

LpxA (Q8RFU2) and LpxD (Q8R6D9) sequences of *F. nucleatum* were retrieved from UniProt [26]. The LpxA and LpxD structures were predicted by homology modelling using the SWISS-MODEL server [27]. The LpxA and LpxD structures of *Escherichia coli* were used as templates for modelling the proteins, respectively. The optimized 3D models were verified using the Procheck [28], ERRAT [29], and ProSA [30] web servers. The predicted models were then compared with their respective template structures and visualized using PyMOL (The PyMOL Molecular Graphics System, Version 2.0, Schrödinger, LLC, New York, NY, USA).

The query coverage, sequence identity, and structure validation properties of the modelled LpxA and LpxD proteins are presented in Table 5. The proteins were assembled as homo-trimers to mimic the biological system for further virtual screening processes (Figure 11 and Figure 12).

### 4.2. Identification of Inhibitors for Library Creation

The inhibitors of small molecules that bind to LpxA and LpxD were searched extensively in various published research works. The known inhibitors of LpxA and LpxD identified in *P. aeruginosa* were considered the starting ligands [13]. Structurally similar small molecules were collected from the PubChem and chEMBL databases [31,32]. The small molecules were verified manually in order to remove compounds that are redundant, compounds other than drugs, mono or dihydrate compounds, compounds that were withdrawn from the market, and compounds with uncertain side effects. The final set of compounds were geometry optimized and converted to PDB structures using Discovery Studios [33].

### 4.3. Virtual Screening of LpxA/LpxD against In-House Compound Library

The LpxA/LpxD protein structures were prepared using Autodock tools. The proteins were prepared by the addition of Gasteiger charges, hydrogen bonds, and merging non-polar hydrogens. The prepared protein structures were saved in PDBQT format. The small molecules for virtual screening were prepared using the Raccoon tool [10,34]. The ligand preparation steps included the addition of Gasteiger charges, polar hydrogens, and merging non-polar hydrogens. The final structures are saved in PDBQT format. The virtual screening of the in-house compound library against LpxA/LpxD proteins was carried out using Autodock Vina [35]. The grid coordinates for docking were set based on the ligand binding sites of the reference ligands from *P. aeruginosa* [13]. The Autodock vina parameters, exhaustiveness, and num_modes were set to 32 and 10, respectively.

### 4.4. ADMET Screening and Activity Spectra

The ligands were screened for toxicity using the DataWarrior tool [36]. The ligands were further filtered based on both toxicological parameters and Lipinski’s rule of five. The screened compounds were further verified for ADME (absorption, distribution, metabolism, and excretion) properties using SwissADME to compute their physicochemical and pharmacokinetics properties [37]. The final compounds that obey ADMET properties were further validated for their biological activity spectrum using the PASS (prediction of activity spectra for substances) server [38]. The PASS server predicts pharmacological effects and biochemical mechanisms on the basis of the structural formula of a substance. The best compounds that show good binding energy and abide by all the criteria were selected for further molecular dynamics (MD) simulation studies.

### 4.5. Molecular Dynamics and Simulation

To understand the structural consequences of ligand binding, a 100 ns molecular dynamics and simulation (MDS) analysis was carried out for apoproteins and ligand-bound protein complexes, respectively. The MDS were performed using GROMACS 2020.6 using the Gromos 54a7 force field [39]. The topology files for the ligands were built using the ATB server [40]. The explicit water model SPC/E (simple point charge/extended) was used to solvate the apoproteins and ligand-bound complexes. The steepest descent algorithm was performed to minimize the system. The apoprotein and ligand-bound complexes were equilibrated in two steps (NVT and NPT). The NVT equilibration for both the systems was performed with constant volume and temperature for 1 ns. Similarly, the NPT equilibration for both the systems was carried out with constant pressure and temperature for 1 ns. A final production MD of 100 ns was carried out for both the apoproteins and ligand-bound complexes. The trajectories of the following MD simulations of the apoproteins and ligand-bound protein complexes were analysed to study the structural parameters, which includes, RMSD (root mean square deviation), RMSF (root mean square fluctuation), Rg (radius of gyration), SASA (solvent accessible surface area), intermolecular hydrogen bonding and PCA (principal component analysis). Further, the apoproteins and protein-ligand complex structures at 0, 50, and 100 ns were also investigated to identify any major structural deviations.

### 4.6. MM/PBSA Based Binding Free Energy

The stable and converged trajectories of the last 20 ns were extracted from the MD simulation of each ligand-bound complex, with 1000 frames recorded at every 50 ps. The trajectories were used to calculate the binding-free energy of the ligand-bound protein complexes using the MM/PBSA (molecular mechanics/Poisson–Boltzmann surface area) approach [41].

Binding-free energy value is calculated using,
∆G=∆Evdw+∆Eele+∆Gpol+∆Gnp

In the above equation, Δ*G* stands for the binding-free energy of the ligand-bound protein complexes, where Δ*E*_vdw_, Δ*E*_ele_, Δ*G*_pol_, and Δ*G*_np_ stands for changes in van der Waals energy, electrostatic energy, polar solvation energy and non-polar solvation energy, respectively.

## 5. Conclusions

In conclusion, targeting the lipid A biosynthetic pathway, particularly focusing on the early acyl transferases LpxA and LpxD, presents a promising approach to combating bacterial infections. Through an in silico methodology, this study has identified Lomefloxacin and Enoxacin as promising drug candidates exhibiting robust activity against Gram-negative bacteria, *Fusobacterium nucleatum*, which is implicated in endodontic infections. These findings not only indicate the potential usage of these drugs for the treatment of endodontic infections but also establish a foundation for the development of novel inhibitors targeting early acyl transferases. Such therapeutic strategies offer significant promise in addressing the challenges posed by bacterial resurgence and antibiotic resistance in clinical settings.

## Figures and Tables

**Figure 1 ijms-25-04239-f001:**
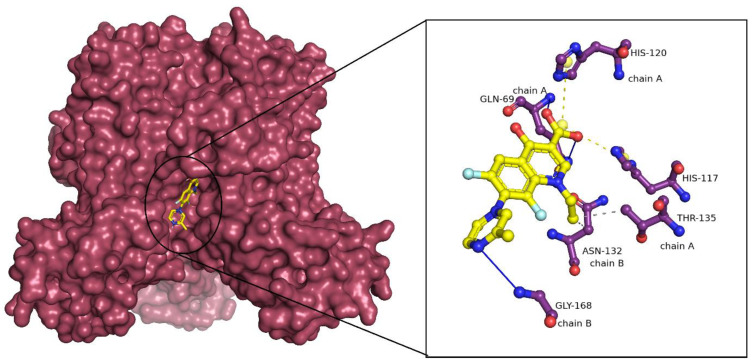
LpxA—Lomefloxacin docked complex. LpxA is represented in surface representation and Lomefloxacin is represented as ball and sticks. The interacting residues of LpxA and Lomefloxacin are represented in ball and sticks format. The conventional hydrogen bonds are shown in blue solid line. Hydrophobic interactions are shown as grey dashed lines and salt bridges are shown as yellow dashed lines.

**Figure 2 ijms-25-04239-f002:**
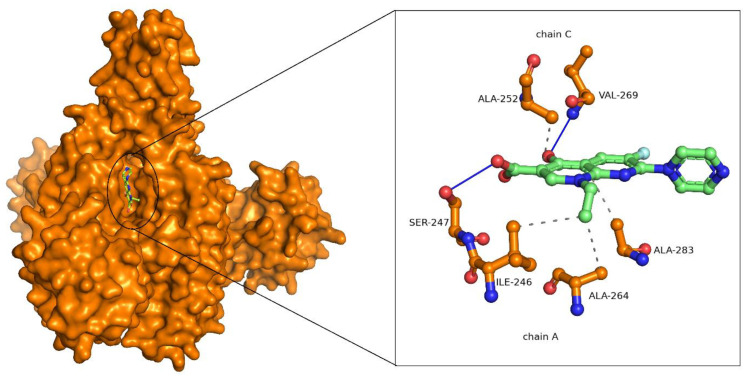
LpxD—Enoxacin docked complex. LpxD is represented in surface representation and Enoxacin is represented as ball and sticks. The interacting residues of LpxD and Enoxacin are represented in ball and sticks format. The conventional hydrogen bonds are shown as a blue solid line. Hydrophobic interactions are shown as grey dashed lines.

**Figure 3 ijms-25-04239-f003:**
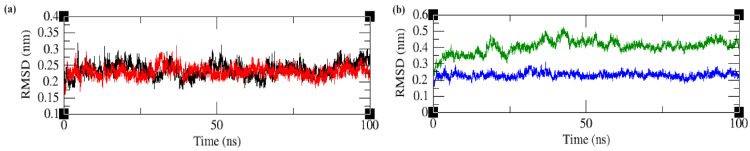
Backbone protein RMSD of (**a**) LpxA system. Black—apo-LpxA and red—LpxA–Lomefloxacin complex. (**b**) LpxD system. Green—apo-LpxD and blue—LpxD-–Enoxacin complex.

**Figure 4 ijms-25-04239-f004:**
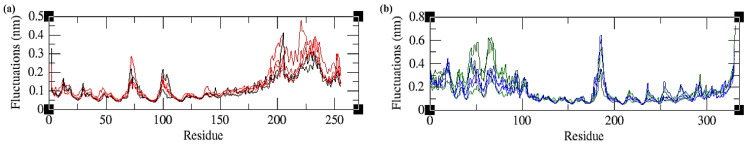
RMSF plot. (**a**) LpxA system. Black—apo-LpxA and red—LpxA–Lomefloxacin complex. (**b**) LpxD system. Green—apo-LpxD and blue—LpxD–Enoxacin complex.

**Figure 5 ijms-25-04239-f005:**
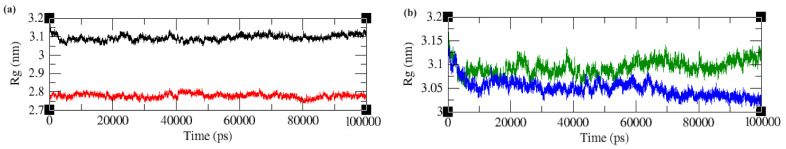
Radius of gyration plot. (**a**) LpxA system. Black—apo-LpxA and red—LpxA–Lomefloxacin complex. (**b**) LpxD system. Green—apo-LpxD and blue—LpxD–Enoxacin complex.

**Figure 6 ijms-25-04239-f006:**
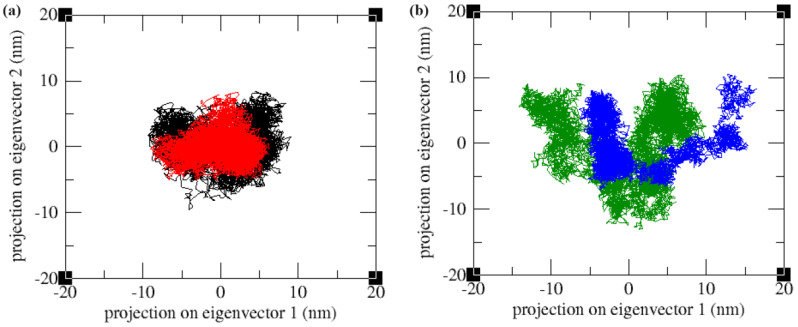
The essential dynamics results of (**a**) the LpxA system. Black—apo-LpxA and red—LpxA–Lomefloxacin complex. (**b**) LpxD system. Green—apo-LpxD and blue—LpxD–Enoxacin complex.

**Figure 7 ijms-25-04239-f007:**
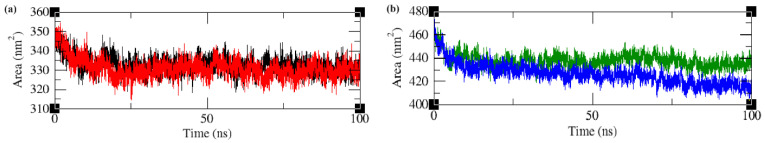
The SASA plots of (**a**) the LpxA system. Black—apo-LpxA and red—LpxA–Lomefloxacin complex. (**b**) LpxD system. Green—apo-LpxD and blue—LpxD–Enoxacin complex.

**Figure 8 ijms-25-04239-f008:**
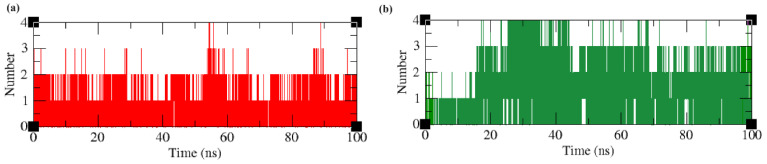
The number of hydrogen bonds obtained from MD simulation. (**a**) red—LpxA–Lomefloxacin complex. (**b**) Green—LpxD–Enoxacin complex.

**Figure 9 ijms-25-04239-f009:**
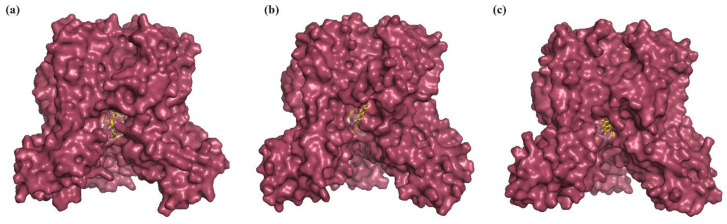
The LpxA–Lomefloxacin complex at (**a**) 0 ns, (**b**) 50 ns, and (**c**) 100 ns. LpxA is represented as a surface, Lomefloxacin is represented as ball and sticks.

**Figure 10 ijms-25-04239-f010:**
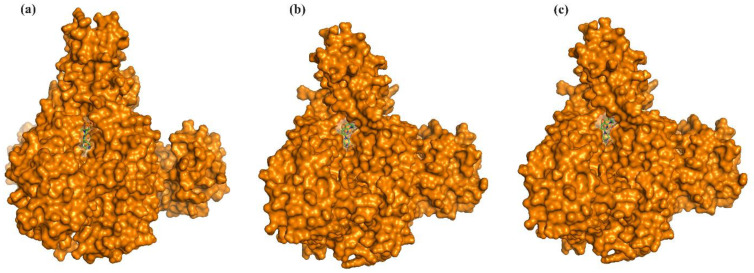
The LpxD–Enoxacin complex at (**a**) 0 ns, (**b**) 50 ns, and (**c**) 100 ns. LpxD is represented as a surface, Enoxacin is represented as ball and sticks.

**Figure 11 ijms-25-04239-f011:**
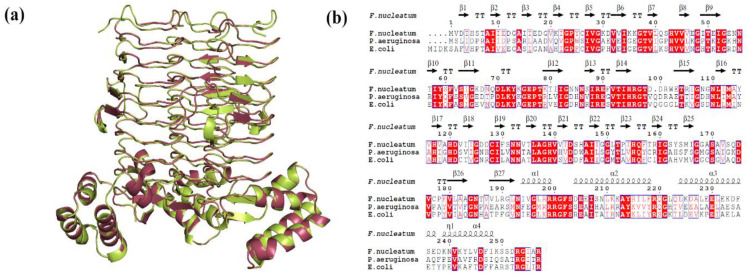
(**a**) Cartoon representation of superimposed structures of modelled LpxA homotrimer over its template; green—*E. coli* LpxA (PDBID: 7OKC) and magenta—*F. nucleatum*. (**b**) Multiple sequence alignment of LpxA sequences of *F. nucleatum. E. coli* and *P. aeruginosa* showing sequence conservation (enclosed in red box) and secondary structural distribution. The secondary structural elements are represented as α- α-helices with the numbers, β—β-strands, η—310-helices, TT—strict β—turns.

**Figure 12 ijms-25-04239-f012:**
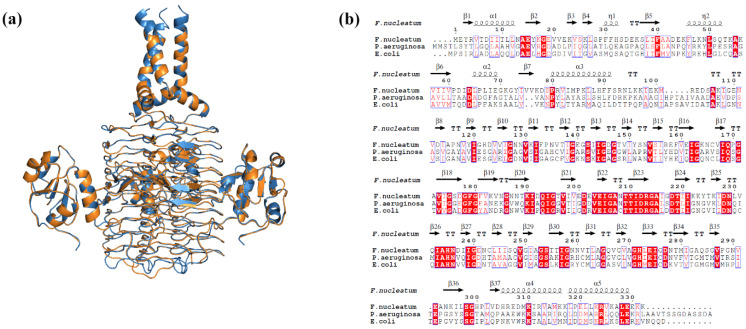
(**a**) Cartoon representation of superimposed structures of modelled LpxD homotrimer over its template; blue—*E. coli* LpxD (PDBID: 3EH0) and orange—*F. nucleatum*. (**b**) Multiple sequence alignment of LpxD sequences of *F. nucleatum. E. coli* and *P. aeruginosa* showing sequence conservation (enclosed in red box) and secondary structural distribution. The secondary structural elements are represented as α- α-helices with the numbers, β—β-strands, η—310-helices, TT—strict β—turns.

**Table 1 ijms-25-04239-t001:** Lipinski’s properties and drug likeness scores of the selected ligands.

Ligand	MW	HBA	HBD	cLogP	DL
Lomefloxacin	353.36	6	3	0.52	5.967
Enoxacin	322.33	6	3	−0.16	36.456

MW: molecular weight; HBA: Hydrogen bond acceptor; HBD: Hydrogen bond donor; cLogP: consensus octanol–water partition coefficient and DL: drug likeness.

**Table 2 ijms-25-04239-t002:** ADME profiles of the selected ligands.

Ligand	GI Absorption	BBB Permeability	TPSA	CYP Inhibitor	PAINS Alert
Lomefloxacin	High	No	76.04	No	0
Enoxacin	High	No	88.40	No	0

GI—gastrointestinal, BBB—blood–brain barrier, TPSA—total polar surface area, CYP—cytochrome P450 and PAINS—pan assay interference compounds.

**Table 3 ijms-25-04239-t003:** Bioactive spectrum scores predicted by PASS server.

Ligand	Activity	Pa	Pi
Lomefloxacin	Anti-infective	0.889	0.004
	Antibacterial	0.624	0.0008
Enoxacin	Antibiotic Naphthyridine-like	0.606	0
	Antibacterial	0.5	0.016

Pa—likelihood of belonging to the class of “Actives”; Pi—likelihood of belonging to the class of “Inactives”.

**Table 4 ijms-25-04239-t004:** MM-PBSA calculations of binding-free energy of the LpxA– and LpxD–ligand complexes.

Complex	ΔG_binding_ *	ΔE_ele_ *	ΔE_vdw_ *	ΔE_ps_ *	ΔE_SASA_ *
LpxA–Lomefloxacin	−95.66 ± 13.20	−13.23 ± 11.06	−159.59 ± 9.65	94.85 ± 9.85	−17.69 ± 1.40
LpxD–Enoxacin	−101.13 ± 11.20	−105.39 ± 16.96	−166.62 ± 7.62	188.01 ± 11.21	−17.13 ± 0.74

* kcal/mol.

**Table 5 ijms-25-04239-t005:** Structure validation statistics for the modelled proteins.

Protein	Template PDB ID	Query Coverage (%)	Identity (%)	Procheck (%)				ERRAT Score ^5^ (%)	ProSA Z Score ^6^	RMSD ^7^ (Å)
				A ^1^	B ^2^	C ^3^	D ^4^			
LpxA	7OKC	98	51.37	88.7	10.0	1.3	0.0	94.33	−7.28	0.110
LpxD	3EH0	98	33.53	83.9	14.8	0.7	0.6	82.32	−6.6	0.129

^1^ Percentage of residues in most favoured region; ^2^ percentage of residues in additionally allowed region; ^3^ percentage of residues in generously allowed region; ^4^ percentage of residues in disallowed region;^5^ structural quality factor; ^6^ overall model quality;^7^ root mean square deviation.

## Data Availability

Data is contained within the article and Appendix A.

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
