# Peer review of "Unveiling Therapeutic Potential: Targeting Fusobacterium nucleatum’s Lipopolysaccharide Biosynthesis for Endodontic Infections—An In Silico Screening Study"

_ijms, 2024, doi:10.3390/ijms25084239_

Round 1

Reviewer 1 Report

Comments and Suggestions for Authors

1. What is the type of the study? please mention in the title

2. Based on the study described report the methods section as per the recommended guidelines and checklist

3. Raw data must be provided for this to validate the results, otherwise can't rule out the merit of the study.

Comments on the Quality of English Language

Moderate English edit to simply English

Author Response

We acknowledge the comments made by  the reviewers.

Answers to reviewer 1 comments

Query 1: What is the type of the study? please mention in the title

Answer 1: Thank you and we acknowledge the comment. The type of the study is in-silico screening. The title is changed to “Unveiling Therapeutic Potential: Targeting Fusobacterium nucleatum's Lipopolysaccharide Biosynthesis for Endodontic Infections – An In-Silico Screening Study”

Query 2: Based on the study described report the methods section as per the recommended guidelines and checklist

Answer 2: The manuscript is revised based on the journals format.

Query 3: Raw data must be provided for this to validate the results, otherwise can't rule out the merit of the study

Answer 3: The raw data is included as supplementary file, File S1.

Reviewer 2 Report

Comments and Suggestions for Authors

The authors suggest that the early acyltransferases LpxA and LpxD, offers a promising avenue for combating bacterial infections. Through an in-silico approach, this study identified Lomefloxacin and Enoxacin as potent drug candidates with strong activity against Gram-negative bacteria, including Fusobacterium nucleatum implicated in endodontic infections. These findings will lay the groundwork for developing novel inhibitors targeting early acyltransferases. Such therapeutic strategies contribute to solve the challenges posed by antibiotic resistance in clinical settings.

In general, the therapeutic approach is novel and relevant, however, the composition of endondontic infections varies substantially. I therefore recommend the authors to further define the target group for this therapeutic strategy. Should it be used also in preventive strategies, or do the microbes in the infected site need to be characterized before initiation of treatment?

Specific points

Line 30-90      The introduction contains several statements that need references.

Line 91             Materials and Methods are fine, however, a graphical abstract would help the reader to fully understand the study.

Line 41-43      The information about Gram negative and Gram positive bacteria is incorrect. It has to be corrected in the revised version.

Line 45-47      Is endodontic infections often composed of several bacterial species in the same root canal/pulp. Add information in the revised version.

Line 48-53      It is crucial for the present study to know how often F. nucleatum is present in endodontic infections. Add this information and a reference to this section.

Line 185          Check space between against and Lipid A in the heading.

Line 186-397 The results are well presented, however, the large amounts of complex results need a result summary. It is not clear how all the presented results connect to each other to full fill the aims of the study.

Line 225-226 Check this sentence: “Lomefloxacin and Enoxacin were further verified for their anti-bacterial activity against published literatures.”

Line 229-232 Are there MIC-values available for the inhibitors?

Line 369-370 Add at least one reference about resistance development.

Line 369-404 The discussion is missing references. Include relevant references in the revised version.

Line 406-414 The conclusions need to be improved. Suggest a shorter more specific conclusion.

Author Response

We acknowledge the comments made by the reviewers.

Answers to reviewer 2 comments

Query 1: The introduction contains several statements that need references

Answer 1: Thank you and we acknowledge the comment. We included additional references in the introduction section of the manuscript as suggested by the reviewer.

Query 2: Materials and Methods are fine, however, a graphical abstract would help the reader to fully understand the study.

Answer 2: We have included a graphical abstract as suggested by the reviewer.

Query 3: The information about Gram negative and Gram positive bacteria is incorrect. It has to be corrected in the revised version.

Answer 3: The list of Gram-positive and Gram-negative bacteria are corrected in the revised manuscript.

Query 4: Is endodontic infections often composed of several bacterial species in the same root canal/pulp. Add information in the revised version.

Answer 4: Yes, endodontic infections commonly consist of multiple bacterial species within the same root canal or pulp. And the information is added as “The microbial diversity within these infections can vary depending on various factors such as the stage of infection, host immune response, and previous treatment history. Several studies have demonstrated the polymicrobial nature of endodontic infections through microbiological analysis techniques such as culture-based methods, polymerase chain reaction (PCR), and next-generation sequencing (NGS) [1–3]”.

Query 5: It is crucial for the present study to know how often F. nucleatum is present in endodontic infections. Add this information and a reference to this section.

Answer 5: As suggested by the reviewer we have included the following information “The frequency of Fusobacterium nucleatum in endodontic infections provide valuable insights into microbial composition and pathogenicity of these infections, influencing diagnostic and therapeutic strategies [7,9,10].” .

Query 6: Check space between against and Lipid A in the heading.

Answer 6: The spacing and typos are corrected in the manuscript.

Query 7: The results are well presented, however, the large amounts of complex results need a result summary. It is not clear how all the presented results connect to each other to full fill the aims of the study.

Answer 7: As suggested by the reviewer, a summary is added at the end of each analysis in the result section of the manuscript for clarity.

Query 8: Check this sentence: “Lomefloxacin and Enoxacin were further verified for their anti-bacterial activity against published literatures.”

Answer 8: The sentence is checked and rephrased to “Lomefloxacin and Enoxacin's antibacterial efficacy was additionally confirmed through examination of published literature” in the manuscript.

Query 9: Are there MIC-values available for the inhibitors?

Answer 9: Unfortunately, no MIC-values are available for the inhibitors against the organism of our study.

Query 10: Add at least one reference about resistance development.

Answer 10: As suggested by the reviewer, the information is added to the result section as “The inappropriate use of antibiotics can lead not only to increased adverse events and healthcare costs but also to the risk of developing resistance [17]. While Lomefloxacin and Enoxacin may exhibit effectiveness in treating periodontal diseases, monitoring for resistance development is essential to ensure continued therapeutic efficacy. The emergence of resistance to Lomefloxacin and Enoxacin in periodontal diseases is a plausible concern. Continuous surveillance is crucial to ensure the sustained effectiveness of these antibiotics.”.

Query 11: The discussion is missing references. Include relevant references in the revised version.

Answer 11: As suggested by the reviewer, the discussion section is revised and relevant references are added.

Query 12: The conclusions need to be improved. Suggest a shorter more specific conclusion.

Answer 12: As suggested by the reviewer, the conclusion section is revised.

Round 2

Reviewer 1 Report

Comments and Suggestions for Authors

There are few sections that need to be connected in this manuscript, although the study objective was to identify the molecules through a computer-aided ligand-based virtual screening approach. Also, authors raised the issue of using Lomefloxacin and Enoxacin.They raised the question of antibiotic resistance in periodontal diseases.

Authors has to connect these with their own findings for example;

Authors found Lomefloxacin and Enoxacin use in endodontic infections but did not clarify how these will be administered practically in clinical scenarios, route administration that is convenient for the patients and the clinicians.

Also when there is a risk of antibiotic resistance due to use in periodontal infections, how is this suitable in managing endodontic infections.

So it is asked to authors to mention that in hypothesis in introduction, and discussion a good discussion with latest citations. The length of the discussion does not matter but has to be connected and help readers understand the rationality and its practical application.

Currently the reporting is biased and does not help any stake holders.

Comments on the Quality of English Language

Moderate English editions required currently it is complex all over the manuscript

Author Response

We acknowledge the comments made by the reviewer.

We express our gratitude to the anonymous reviewers and the editor for their valuable contributions in improving our manuscript.

Answers to reviewer 1 comments

Comment 1: Authors raised the issue of using Lomefloxacin and Enoxacin. They raised the question of antibiotic resistance in periodontal diseases.

Answer 1: Thank you for your valuable feedback. We have incorporated an introduction section to provide an overview of antibiotic resistance. Additionally, we have addressed the issue of antibiotic resistance and included strategies for tackling it in the discussion part of the manuscript.

Comment 2: Authors found Lomefloxacin and Enoxacin use in endodontic infections but did not clarify how these will be administered practically in clinical scenarios, route administration that is convenient for the patients and the clinicians.

Answer 2: Thank you for your comment. We revised the manuscript by the providing details in the discussion section.

Comment 3: Also when there is a risk of antibiotic resistance due to use in periodontal infections, how is this suitable in managing endodontic infections. So it is asked to authors to mention that in hypothesis in introduction, and discussion a good discussion with latest citations.

Answer 3: Thank you for your comment. We understand the importance of addressing the risk of antibiotic resistance associated with the use of Lomefloxacin and Enoxacin in periodontal infections and its implications for their suitability in managing endodontic infections. We revised the introduction to include this consideration, and ensured that the discussion section contains strategies for tackling resistance along with citations.

Reviewer 2 Report

Comments and Suggestions for Authors

The revised manuscript is substantially improved I  have no further comments to add.

Author Response

We acknowledge the comments made by the reviewer.

We express our gratitude to the anonymous reviewers and the editor for their valuable contributions in improving our manuscript.

Comments: The revised manuscript is substantially improved I have no further comments to add.

Answer: We thanks to reviewer 2.  There is no comments.